# Sorption Preconcentration and Analytical Determination of Cu, Zr and Hf in Waste Samarium–Cobalt Magnet Samples

**DOI:** 10.3390/molecules27165275

**Published:** 2022-08-18

**Authors:** Alexandra Alexandrovna Arkhipenko, Kseniya Vadimovna Petrova, Vasilisa Borisovna Baranovskaya

**Affiliations:** Kurnakov Institute of General and Inorganic Chemistry, Russian Academy of Sciences, Leninskij Prospect, 31, Moscow 119991, Russia

**Keywords:** sorption, recycling, ICP-OES, S- and N-containing sorbent, preconcentration, waste Sm-Co magnets

## Abstract

We developed a method of sorption determination via the atomic emission of Cu, Zr and Hf metals in the waste of samarium–cobalt magnets. This method was based on the preconcentration of impurities using S- and N-containing heterochain sorbents, with further determination of the analytes via inductively coupled plasma atomic emission spectrometry (ICP-OES). Different sorbents such as PED (polyethelendiamine), TDA (polythiodimethanamine), PhED (*N*-phenylpolyethediamine) and PTE (polythioether) were tested for Ti, Cu, Zr, Nb and Hf extraction. The PTE sorbent ensured the maximum extraction of the analytes (recovery 60% for Ti, 80% for Nb, 95–100% for Cu, Zr and Hf) and thus was selected for further research. Additionally, various acidities of chloride solution (0.01–1 M HCl) were investigated for metal sorption. Under the optimised sorption conditions, trace impurities of Cu, Zr and Hf were determined using ICP-OES with a relative standard deviation of less than 5%. The obtained results were confirmed by the added–found method and cross-method experiments. The detection limits (DLs) were 1.5, 2, 0.15, 2 and 0.75 µg/L for Ti, Cu, Zr, Nb and Hf, respectively. The proposed method can be successfully used for the determination of various microelements in other waste REE-magnetic materials.

## 1. Introduction

The modern world is highly dependent on its resources. The depletion of natural resources forces humanity to deal with the problem of secondary resources. Whilst metal-containing secondary raw materials are diverse, their valuable component content is limited when compared to that of natural raw materials. Another aspect of the problem is related to the fact that over the years, a large amount of metal-containing waste has accumulated, causing enormous damage to the environment. This waste continues to grow. Therefore, the problems of collection, chemical analysis and recycling of metal-containing waste must be dealt with both from an economic and environmental point of view.

It is known that spent rare earths (RE) contain magnets (NdFeB and SmCo alloys) that are examples of raw secondary-metal-containing materials with regard to rare, non-ferrous, high-melting metals [1,2,3,4,5]. The development and/or selection of processing (in our case extraction) technology of waste magnetic materials significantly depends on their chemical analysis.

The methods used to analyse rare-earth-containing magnets include inductively coupled plasma atomic emission spectrometry (ICP-OES) [6,7,8,9], X-ray fluorescence (XRF) [10] and inductively coupled plasma mass spectrometry (ICP-MS) [3,11]. However, most of the research is focused on the determination of the major components (Sm, Co, Fe, Nd etc.) in RE-based magnets, whilst the analysis of magnetic waste is dramatically overlooked. Waste magnetic materials contain non-stereotypical elements, making analysis much more challenging than that of alloys with the predicted composition. For example, waste Sm-Co magnets can contain rare and non-ferrous metals, namely Cu, Zr, Nb, Ti, Hf, etc. Their determinations are relevant in terms of their secondary uses, in particular for microelectronics.

However, few articles have been published on the determination of impurities in RE-based magnets and their waste [8,9,10,11]. Among them, part of the work is devoted to the ICP-OES analysis of NdFeB magnetic materials [8,9]. Only a few articles on the spectral analysis of Sm-Co magnets and their waste have been found [10,11,12]. In the literature, [10] ICP-OES was used to determine Sm, Co, Cu, Fe, Zr, Ce, Hf, Mn, Fe and Y in the range of n*10^−2^–62 wt, %. The authors [11] developed a spectral method for the determination of Sm, Co, Cu, Fe and Zr in the concentration range of n*10^−2^–20 wt, %. In our previous work [12], the ICP-OES method allowed us to determine the trace and matrix components of waste Sm-Co magnets with the limits of quantification (LOQs) for most elements being n·10^−4^ wt. %. Nevertheless, the determination of trace elements (below 10^−4^ wt. %) is essential for rational recycling schemes with regard to waste Sm-Co magnets.

The large number of emission lines from matrix elements make the determination of trace elements in waste Sm-Co magnets complicated. Thus, the application of the ICP-OES method becomes challenging. That is why sample pretreatment methods, such as extraction and sorption, are often necessary prior to analysis [4,8,13,14,15,16,17,18,19,20]. Most published articles are devoted to the recovery of the main components and rare earth impurities from waste Sm-Co magnets.

One of the universal techniques of element preconcentration in such a complex, multicomponent and non-standard material as secondary rare-metal-containing raw material is sorption preconcentration based on S- and N–containing sorbents [21,22,23,24]. These sorbents have been successfully used for the preconcentration of platinum and toxic elements from different kinds of secondary raw materials. The aim of this work is to investigate the capabilities of these sorbents for the sorption of rare and non-ferrous metals (Ti, Cu, Zr, Nb and Hf) from waste Sm-Co magnets. In this work, four types of S- and N–containing sorbents were studied and the experimental parameters of sorption preconcentration were examined in detail using reference materials (RMs) of waste Sm-Co magnets. On the basis of experimental results, a novel method of coupling sorption preconcentration, using the PTE thioether sorbent for ICP-OES, was developed for the simultaneous determination of trace Cu, Ti, Zr, Hf and Nb in waste Sm-Co magnets, with satisfactory results.

## 2. Materials and Methods

### 2.1. Apparatus

A simultaneous Thermo Scientific iCAP PRO XP ICP-OES (Thermo Fisher Scientific Inc, Waltham, MA, USA) with a vertical torch, a purged echelle polychromator and a charge injection device (CID) array detector were used. A standard sample introduction kit suitable for aqueous samples consisting of a glass cyclonic spray chamber, a SeaSpray glass nebuliser and a quartz glass duo torch, among other components, was also used. The dual-view torch provided versatility in viewing configurations, with a radial view for complex matrices and an axial view for high sensitivity. Instrumental operating conditions are summarised in Table 1.

An inductively coupled plasma mass spectrometer, XSeries II, developed by Thermo Techno Scientific (Waltham, MA, USA), was used for comparative analysis and to verify the accuracy of the results. Instrumental operating conditions are summarised in Table 2.

The sorbents were characterised using a PerkinElmer Spectrum 65 Fourier transform infrared (FT-IR) spectrometer (PerkinElmer Instruments, Norwalk, CT, USA.). The IR spectra were recorded in the 400–4000 cm^−1^ region in the attenuated total reflection (ATR).

The NMR spectra were measured on a Bruker Avance 400 spectrometer (Bruker Corporation, Billerica, MA, USA) at room temperature; the chemical shifts δ were measured in ppm with respect to solvent (^1^H: CDCl3, δ = 7.28 ppm; DMSO-*d_6_*: δ = 2.50 ppm; ^13^C: CDCl_3_, δ = 77.2 ppm; DMSO-*d_6_*: δ = 39.5 ppm). The structures of synthesised compounds were elucidated with the aid of ^1^H and ^13^C spectroscopy.

The pH was measured with pH-meter model I-160 MI (OOO Izmeritel’naja tehnika, Moscow, Russia).

The sorption systems were heated using an ultrasonic bath (without switching on ultrasound), ODA-MH20 (OdaServis, Moscow, Russia).

### 2.2. Reagents and Materials

High-purity HNO_3_ was used in the dissolutions. Deionised water with a resistivity of 18.2 MΩ cm at 25 °C was used for all dissolutions and dilutions.

Aqueous ICP-OES calibration solutions in 5% HNO_3_ were prepared; they were obtained from High-Purity Standards (North Charleston, SC, USA).

Calibration solutions were prepared in a concentration range selected to cover the full range of trace and matrix elements in waste samarium–cobalt magnets: 0.01–10 mg/L for Ti, Cu, Nb, Zr and Hf; 500–1500 mg/L for Sm and Co.

Selectivity was studied using standard solution from High-Purity Standard (North Charleston, SC, USA) containing 100 mg/L Al, As, B, Ba, Be, Bi, Ca, Cd, Co, Cr, Cu, In, Fe, Ga, Hf, K, Li, Mg, Mn, Na, Nb, Ni, Pb, Sc, Se, Si, Sn, Sr, Ta, Te, Ti, Tl, V, Zn and Zr.

The selection of analytical lines was carried out by sampling 1000 mg/L solutions of Sm and Co, prepared from 1 and 10 g/L standard stock solutions from High-Purity Standards (USA) and pure solutions containing 1 mg/L of Ti, Hf, Nb, Cu and Zr.

Argon of 99.996 purity grade was used for the ICP-OES and ICP-MS measurements.

The reference materials (RMs) of waste magnets (RM 1, RM 2, RM 3, RM 4) were developed at the Federal State Research and Development Institute of Rare Metal Industry “Giredmet” (Moscow, Russia). The chemical compositions of the RMs were determined using ICP-MS. The certified concentrations of the RMs are indicated in the Table 3.

For the extraction and concentration of Ti, Cu, Zr, Nb and Hf, the developed polymer S-containing and S- and N-containing sorbents were used. The sorbents were synthesised at the Federal State Research and Development Institute of Rare Metal Industry “Giredmet” (Russia). The sorbent structure is shown in Figure 1. The sorbent PTE is polythioether, PED is polyethelendiamine, TDA is polythiodimethanamine and PhED is *N*-phenylpolyethediamine. These sorbents are promising, convenient to use, easily soluble in mineral acids, cheap to manufacture and easily synthesised in the laboratory.

To increase the sorption efficiency of the sorbent, aqueous ammonia (Himmed, Russia), ethylenediaminetetraacetic acid (EDTA) (ACS reagent, Merck, Germany), salicylic acid (ACS reagent, Merck, Germany), acetamidine (ReagentPlus, Merck, Germany), methylamine (ReagentPlus, Merck, Germany), hydroxylamine (ReagentPlus, Merck, Germany), trimethylamine (ReagentPlus, Merck, Germany) and dithiothreitol (ReagentPlus, Merck, Germany) were used.

### 2.3. Synthesis of Sorbents

The synthetic scheme of the PTE sorbent is shown in Figure 2a. Paraformaldehyde was saturated with gaseous hydrogen sulphide at 30 °C [23]. Synthesis is also possible in the presence of Na_2_S instead of H_2_S under ultrasonication [25].

For the synthesis of PED (Figure 2b) an aqueous solution of formaldehyde was introduced into a three-neck flask equipped with a thermometer, a mechanical stirrer and an additional funnel. The solution of ethylenediamine was added dropwise. The temperature of the reaction mixture was monitored and did not exceed 20 °C. After that, the mixture was left to settle for 20 h and then exposed to gaseous hydrogen sulphide at 70 °C. The reaction was carried out to constant weight [26]. In our case, the functional groups of the sorbent were deposited on the anion exchanger.

The synthesis of TDA (Figure 2c) was similar to the synthesis of the Ed sorbent, but formaldehyde was added, together with hydrogen sulphide, at 90 °C.

The synthesis of PhED (Figure 2d) differed from the previous ones in that n-toluidine was used in the synthesis instead of ethylenediamine.

### 2.4. Sample Preparation

A 0.5 g sample of waste samarium–cobalt magnet was dissolved in 10 mL of high-purity HCl at 150 °C for 30 min, shaking every 5 min. After cooling, the solution was brought up to 100 mL with deionised water. The common components of waste samarium–cobalt magnets can easily be dissolved in such conditions.

### 2.5. Sorption Procedure

Sorption concentration was carried out to separate the determined elements from matrix components. A 10 mL sample solution containing target ions was placed in a 15 mL test tube, with an acidity of 0.1 M (HCl). Then, 100 mg of sorbent was added to the test tube and the mixture of sorbent and sample was heated using the ultrasonic bath to 60 °C for 30 min. The mixture of sorbent and sample was heated to increase the recovery. The resulting solution was filtered through a quantitative filter paper “white ribbon” (pore diameter of 8–12 µm) in a 15 mL test tube, washing the tube walls with sorbent two or three times with deionised water. The filter was washed with water, collecting sorbent in the centre of the filter. The deposit was washed away from the filter in a flask, dissolved in 1 mL of nitric acid during heating and diluted to 5 mL. We established that all the sorbent was washed from of the filter: after completely washing away the deposit, the filter was washed with concentrated nitric acid, and ICP-OES analysis demonstrated that the concentration of analytes on the filter was below the limit of detection. Finally, the resulting solution was analysed using ICP-OES.

## 3. Results and Discussion

### 3.1. Characterisation of Sorbents

The FTIR spectra for the sorbents are shown in Figure 3. It can be seen that the PTE spectrum confirms it to have a polythioether structure. The absorption peaks in the range 1400–1000 cm^−1^ characterise its presence in the –CH_2_-group. The presence of grafted S is confirmed by the appearance of C-S and C-S-C bonds from 770 to 550 cm^−1^.

The sorbent PED, like TDA and PhED, has a branched complex structure. Therefore, its absorption peaks in the spectrum can be split. The area 1700–1450 cm^−1^ in PED indicates the presence of numerous methyl groups. The peak at 1484 cm^−1^ indicates the presence of a –CH_2_ group in the sorbent. The spectrum shows the bands corresponding to N–C bonds in the range 1102–986 cm^−1^. The presence of C-S bonds in PED is confirmed by the intense broad band of peaks in the range 706–420 cm^−1^.

The spectrum of TDA suggests that the area 1238–856 cm^−1^ is attributable to -CH_3_ and -CH_2_ groups. The peak located at 784 cm^−1^ is the characteristic frequencies of the N–C bond. The band covering values of 500 cm^−1^ and below is the characteristic frequencies of the S–C bond.

The PhED sorbent differs from the others because of the presence of a phenyl group. This is illustrated by the band occupying the range 3056–2688 cm^−1^. The peaks at 1662 and 1514 cm^−1^ indicate the –CH_3_ and –CH_2_ groups. The band extending from 1332 to 1080 cm^−1^ represents the characteristic frequencies of the N–C bond. The N–Ph bond is characterised by the 873–640 cm^−1^ band. The presence of the C–S grafted bond can be seen on the 562–444 cm^−1^ band.

Thus, it can be assumed that all four sorbents correspond to the declared structure [27,28].

Analysis of the ^1^H NMR spectrum of PTE (Figure 4a) allows us to determine the following key points. Singlets at 3.93 and 3.92 Hz mean the potential presence of methyl groups in the sorbent molecule, or refer to the many CH_2_ groups that are split on top of each other. There are no methyl groups in PTE, therefore, it can be assumed that these signals are CH_2_ groups. In addition, in this spectrum we observe a quadruplet and a doublet at 5.89 and 4.70 Hz, respectively. These signals are integrated to 2 and 4, respectively, which tells us that these are the signals of hydrogens with CH_2_ groups that are closer to the end of the molecule.

The spectrum of the PED (Figure 4b) sorbent is the most difficult to analyse as the sorbent was hardly soluble due to the fact that it was applied to an anion exchange resin. Thus, in using DMSO-*d_6_* as a deuterium solvent, only a slight degree of solubility was achieved, and the signals that were the signals of the sorbent were slightly above the background level. However, taking into account the polymeric structure of the compound and the possibility for hydrogen atoms to split on the sulphur heteroatom, the neighbouring CH_2_ functionalities and the nitrogen atom, it can be assumed that the NMR spectrum corresponds to the given structure.

When studying the ^1^H NMR spectrum of TDA (Figure 4c), we can observe some characteristic signals. We observe a singlet at 3.46 Hz, which is characteristic of the methyl(s) groups located in close proximity to the nitrogen atom. Since the structure of the sorbent contains the dimethylamino function, it is reasonable to assume that this singlet corresponds to this function and integrate it to 6 protons. Then, on the NMR spectrum, we can observe two broadened multiplets (2.60 and 2.36 Hz), which are integrated with respect to the known singlet by 2 protons each, respectively; because of this, we have the right to assume that these signals correspond to CH_2_ groups located in the molecule. The broadness of these signals indicates their polymeric nature, and the magnitude of the shifts indicates the immediate proximity of the sulphur atom.

When studying the spectrum of PfED (Figure 4d), one sees a number of signals characteristic of hydrogen atoms contained in aromatic structures. We are referring to the 5 protons of the phenyl groups, of which there are 3 pcs in the sorbent monomer. The range of signals from 6.53 to 7.56 Hz corresponds to aromatic signals. The signals in the range from 2 to 5.5 Hz correspond to the signals of CH_2_ groups of the polymer chain. Moreover, after integrating in accordance with integer values, we can make a conclusion about the approximate ratio of atoms to each other n:m as 3:2. According to the given proton spectrum, one-valued correspondence of the expected structure to its NMR spectrum can be asserted.

### 3.2. Sorption without Additional Reagents

Table 4 shows the results of the sorption extraction of impurities from SmCo-solutions without the introduction of additional reagents. It can be seen that the sorption capacity of the sorbent was not enough to completely extract Ti, Cu, Zr, Nb and Hf from SmCo-solutions.

### 3.3. Sorption in an Acidic Environment

The ability of the sorbents to promote the recovery of the analytes was studied in hydrochloric acid and nitric acid solutions. This was justified by the increased sorption capacity of sorbents in acidic solutions [21,22]. In addition, acid is used to dissolve magnets. Table 5 shows the results of the sorption of Ti, Cu, Zr, Nb and Hf elements from hydrochloric acid and nitric acid solutions. These data demonstrate that sorption from hydrochloric acid solution was slightly higher than from nitric acid solution. This can be explained by the formation of a stable metal cation for further bonding with the sorbent.

### 3.4. Introduction of Additional Sorption Reagents

To increase the recoveries of analytes, various sorption reagents were used that could promote the formation of complexes in the element-sorbent-chloride-anion system. Reagents were selected for the study due to their ability to form complexes: EDTA [29,30,31,32,33], ammonia [34,35,36,37,38], acetamidine [39,40], hydroxylamine [41,42], methylamine [37,38,43,44], trimethylamine [45,46,47], salicylic acid [48,49,50,51] and dithiothreitol [52]. Figure 5 shows that the highest recovery was achieved using the PTE sorbent. This sorbent was chosen for further research. EDTA was chosen for the group extraction of Cu, Zr and Hf and ammonia was chosen for the group extraction of Ti, Nb, Zr and Hf for their ability to promote sorption extraction.

The proposed mechanism of reaction with EDTA (a) and ammonia (b) is shown in Figure 6. The metal atom forms a more stable complex with sulphur atoms than the nitrogen atom in the sorbent and additional reagent.

### 3.5. The Effect of Reagent Concentration

To more accurately determine impurities in a solution, it is necessary to choose the correct concentration of the sorption reagents that are to be introduced. Too low a concentration will be insufficient for the formation of bonds between the metal and the sorbent; too high a concentration will interfere with the determination performed by the ICP-OES method due to matrix effects. The results of varying the concentrations of ammonia and EDTA for PTE are presented in Figure 7. The data indicate that 50 mg/L EDTA was sufficient for the bulk extraction of copper, zirconium and hafnium. A lower concentration would most likely have been insufficient for the formation of metal–sorbent bonds, while a higher concentration would have reduced the detectable concentration, probably due to spectral overlaps.

For the complete extraction of zirconium and hafnium by the sorbent phase, the highest concentration of ammonia, 2500 mg/L, was required. A lower concentration was insufficient for the formation of sorbent–metal bonds.

### 3.6. The Effect of Sample Acidity

The acidity of a sample solution greatly influences the existing form of its analytes. In addition, too high an acidity value can destroy the functional groups of the sorbent. Therefore, the effect of sample acidity on recovery was examined at different concentrations of hydrochloric acid in solution. It can be seen from Figure 8 that the recovery of the analytes decreased with the increase in acidity of the sample solution from 0.01 mol/L (pH 1.86) to 1 mol/L (pH 1.55) HCl, and that quantitative recovery (95–100%) of the analytes was obtained at 0.01–0.1 mol/L HCl. Thus, 0.1 mol/L HCl of the sample solution was used in the current study, because this amount of acid was sufficient to dissolve the sample.

### 3.7. The Study of Kinetics

Sorption is influenced by the time of contact between the phases of the sorbent and the sample solution. Sorption equilibrium was established in the first 30 min of phase contact; therefore, this time was sufficient for the complete recovery of target elements, as can be seen from Figure 9. The maximum recovery of analytes in the shortest time was achieved within 30 min.

### 3.8. The Effect of Temperature

The temperature of the system affects the establishment of sorption, as it directly affects the formation of metal–sorbent bonds and finally the kinetics of the sorption process. The sorption process is presumably endothermic, so heating accelerates the extraction process. The results of varying temperature are shown in Figure 10. It was established that a temperature of 60 °C was sufficient to establish sorption equilibrium. A further increase in temperature would have been inefficient, possibly leading to the destruction of functional groups.

### 3.9. Influence of Adsorbent Dose

To activate the optimum amount of sorbent necessary for the quantitative recoveries of analytes, different masses of PTE were tested from 0.005 to 0.3 g. The results are shown in Figure 11. The dependence means that quantitative recoveries of Cu, Zr and Hf could be obtained using amounts of PTE ranging from 0.1 to 0.2 g. The maximum recovery of Nb and Ti corresponds to the same mass range. Thus, 0.1 g of PET was used in following experiments. A large mass of the sorbent (>0.2 g) during dissolution affected the resulting spectral signal. Figure 10 shows a decrease in recovery with a sorbent mass of 0.3 g.

### 3.10. The Matrix Effect

The interfering influence of matrix elements is presented in Table 6. It was found that Sm affected the extraction of copper and that Co had an effect on the extraction of titanium, zirconium and hafnium.

To assess the applicability of this method, different concentrations of potentially interfering ions were individually added to the sample solution, which contained 1 ppm of analytes, and their effects on the recovery of the analytes were investigated under optimum conditions. It was found that Al, As, B, Ba, Be, Bi, Ca, Cd, Co, Cr, In, Fe, Ga, K, Li, Mg, Mn, Na, Ni, Pb, Sc, Se, Si, Sn, Sr, Ta, Te, Ti, Tl, V and Zn at a concentration of 1 ppm did not affect recovery.

### 3.11. Analytical Performance

The analytical parameters of the ICP-OES method, such as detection limits, precision, correlation coefficients and linear range, were evaluated under optimum conditions. The detection limits (DLs), defined as the concentration of the analytes equal to three times the standard deviation from nine replicate detections of the blanks, were 1.5, 2, 0.15, 2 and 0.75 µg/L, for Ti, Cu, Zr, Nb and Hf, respectively.

The relative standard deviations (RSDs) were 1.0%, 1.1%, 0.2%, 0.3% and 0.5% for Ti, Cu, Zr, Nb and Hf (*n* = 10, c = 0.8 mg/L), respectively.

ICP-OES has wide dynamic range. The linear range of calibration was over four orders of magnitude, with a correlation coefficient in excess of 0.998. Rectilinear calibration lines were confirmed for 0.01 to 10 mg/L of minor additives, Ti, Cu, Nb, Zr and Hf.

When studying the effect of analyte concentration on recovery, we found that the maximum recoverable concentration of analyte was 7 mg/L. With increasing concentration, recovery decreased. The minimum concentration of recovery is limited by the limit of detection by ICP-OES.

### 3.12. Analysis Application

The accuracy of this method was examined by the determination of Cu, Zr and Hf in waste SmCo reference materials (RM). Reliability was checked by spiking the sample. The results, along with recovery data for the spiked sample, are presented in Table 7.

To further validate this method, the ICP-MS was used. The results, shown in Table 8, indicated that the values obtained closely matched the ICP-MS values. The inter-laboratory precision (I(TO)) and repeatability limit (r) of the results were calculated in accordance with ISO 5725. For most elements, the RSDs of the obtained results did not exceed 6.4%.

## 4. Conclusions

In summary, S- and N-containing sorbents were studied and used as adsorbents for the preconcentration of Ti, Cu, Zr, Nb and Hf in waste Sm-Co magnets prior to ICP-OES determination. The structure and sorption capacity of the sorbents were investigated and the concentration conditions were optimised. We found that the PTE sorbent was the most effective for the preconcentration of the analytes (Ti, Cu, Zr, Nb and Hf).

Cu, Zr and Hf were quantitatively recovered (100 ± 3, 98 ± 4, 95 ± 4% respectively) at 50 mg/L EDTA using the PTE sorbent; Zr and Hf were quantitatively recovered at 2500 mg/L ammonia using the PTE sorbent. The maximum extraction of Ti was 60 ± 5% and that of Nb was 82 ± 3% in 2500 mg/L ammonia in both cases using PTE sorbent. It was established that the most acceptable concentration of hydrochloric acid was 0.1 mol/L. For maximum recovery, 30 min at 60 °C was sufficient.

Other sorbents were less effective. Sorption by the TDA sorbent was 94 ± 2, 91 ± 3, 85 ± 3, 75 ± 3 and 46 ± 2% for Cu, Hf, Zr, Nb and Ti respectively. Sorption by the PED sorbent was 93 ± 1, 90 ± 2, 85 ± 2, 85 ± 3 and 51 ± 1% for Cu, Hf, Zr, Nb and Ti respectively. Sorption by the PhED sorbent was 100 ± 5, 88 ± 3, 88 ± 4, 76 ± 3 and 53 ± 2% for Cu, Hf, Zr, Nb and Ti respectively.

This work reveals the potential application of S- and N-containing sorbents for the preconcentration, separation and analysis of trace metals in REE-magnet products for their recycling. The developed method can be used for analytical control of waste Sm-Co magnets during the recycling process using extraction with deep eutectic solvents based on trioctylphosphine.

## Figures and Tables

**Figure 1 molecules-27-05275-f001:**
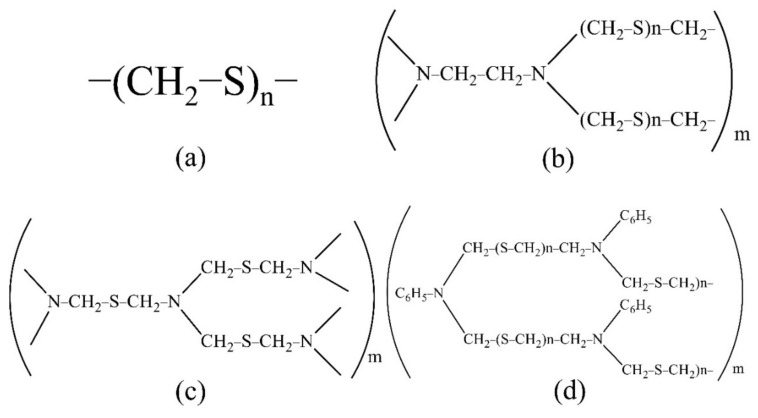
The structure of the sorbents studied ((**a**): PTE, (**b**): PED, (**c**): TDA, (**d**): PhED).

**Figure 2 molecules-27-05275-f002:**
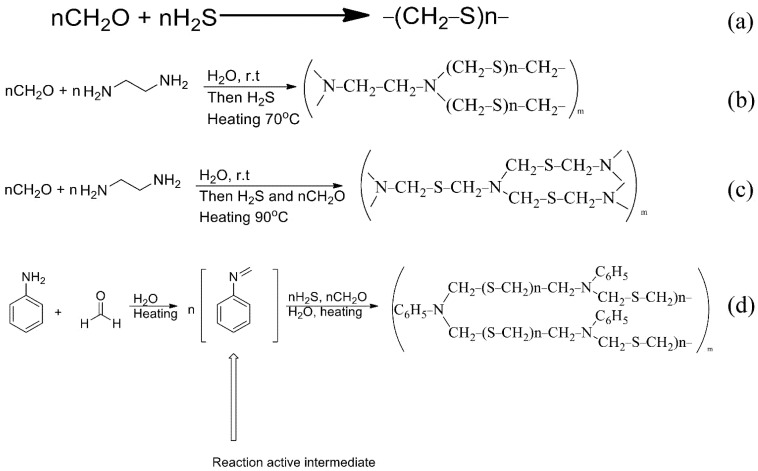
The schemes of reactions of synthesis of sorbents ((**a**): PTE, (**b**): PED, (**c**): TDA, (**d**): PhED).

**Figure 3 molecules-27-05275-f003:**
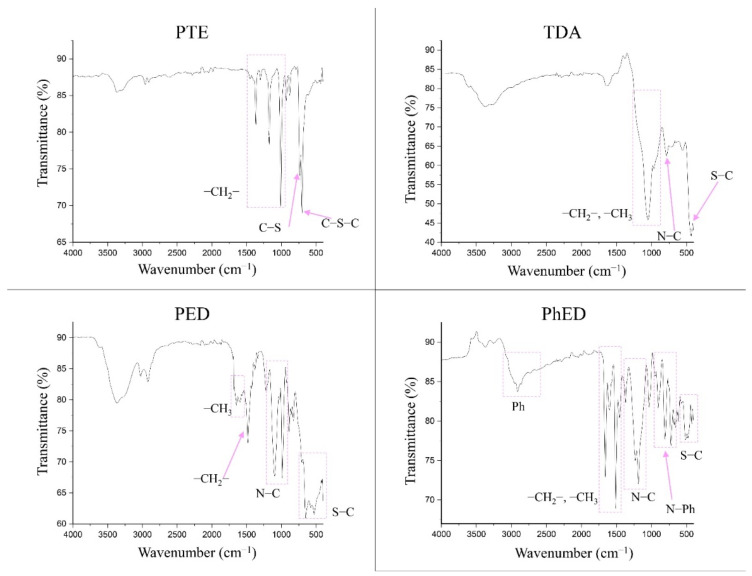
IR spectra of sorbents.

**Figure 4 molecules-27-05275-f004:**
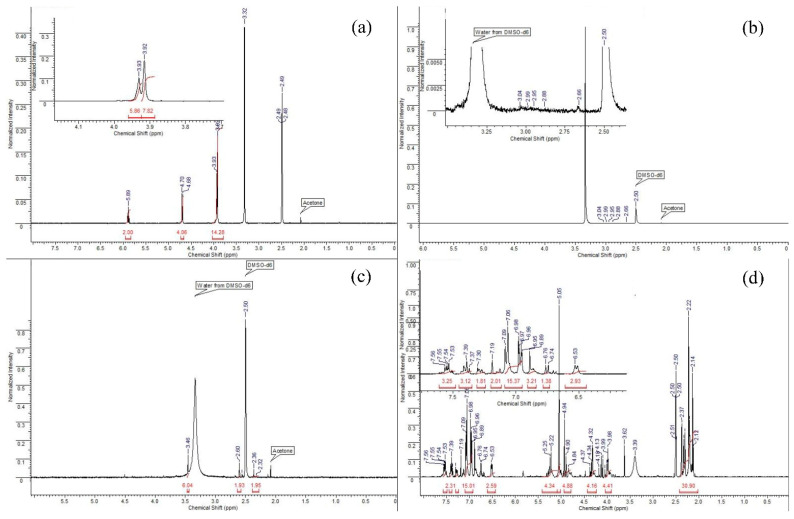
The ^1^H NMR spectrum of sorbents ((**a**): PTE, (**b**): PED, (**c**): TDA, (**d**): PhED).

**Figure 5 molecules-27-05275-f005:**
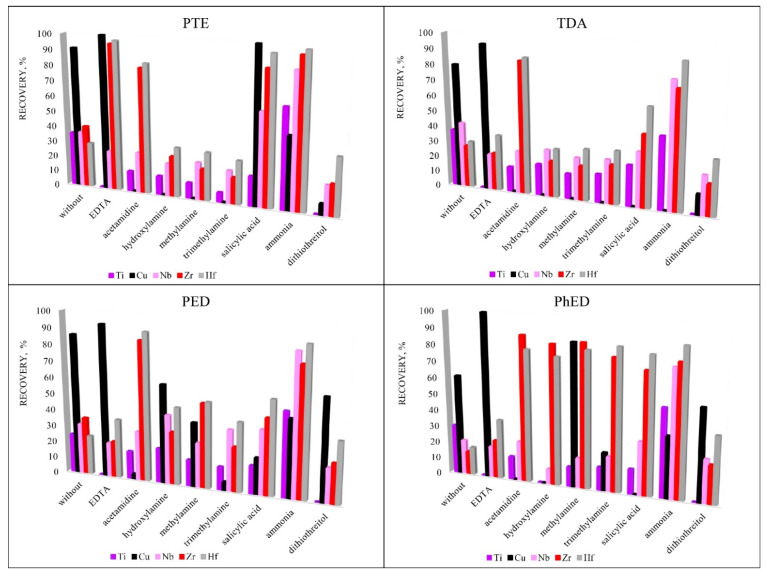
The influence of additional reagents on recovery. Sorption conditions: 0.1 mol/L HCl, c_additional reagents_ = 5 mg/L, m_sorbent_ = 0.1 g, V _liquid phase_ = 10 mL, t = 1 h, T = 60 °C, C_analytes_ = 1 mg/L.

**Figure 6 molecules-27-05275-f006:**
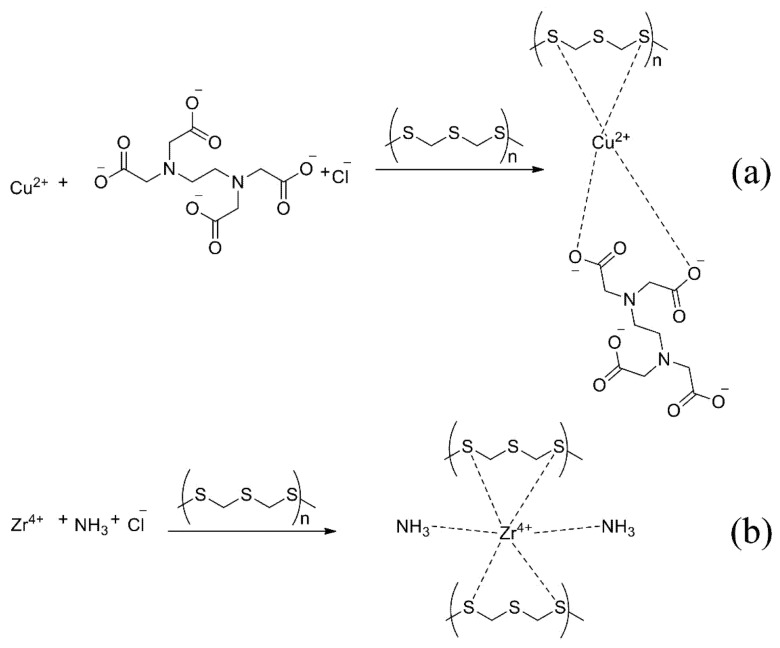
The proposed mechanism of sorption with EDTA (**a**) and ammonia (**b**).

**Figure 7 molecules-27-05275-f007:**
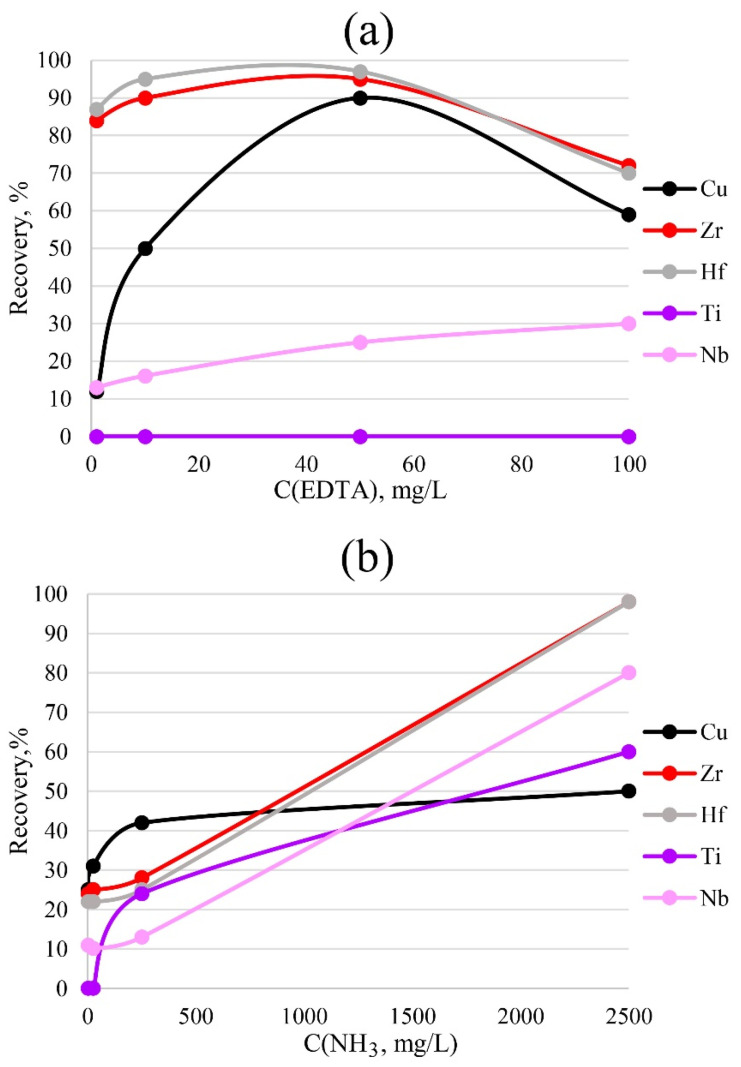
Dependence of recovery on the concentrations of EDTA (**a**) and ammonia (**b**). Sorption conditions: 0.1 mol/L HCl, m_PTE_ = 0.1 g, V _liquid phase_ = 10 mL, t = 1 h, T = 60 °C, C_analytes_ = 1 mg/L.

**Figure 8 molecules-27-05275-f008:**
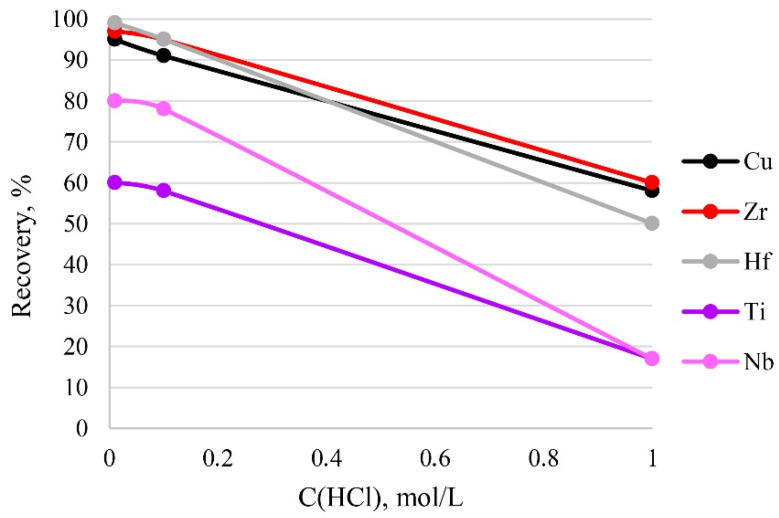
The dependence of recovery on acidity. Sorption conditions: C_EDTA/NH3_ = 50/2500 mg/L, m_PTE_ = 0.1 g, V _liquid phase_ = 10 mL, t = 1 h, T = 60 °C, C_analytes_ = 1 mg/L.

**Figure 9 molecules-27-05275-f009:**
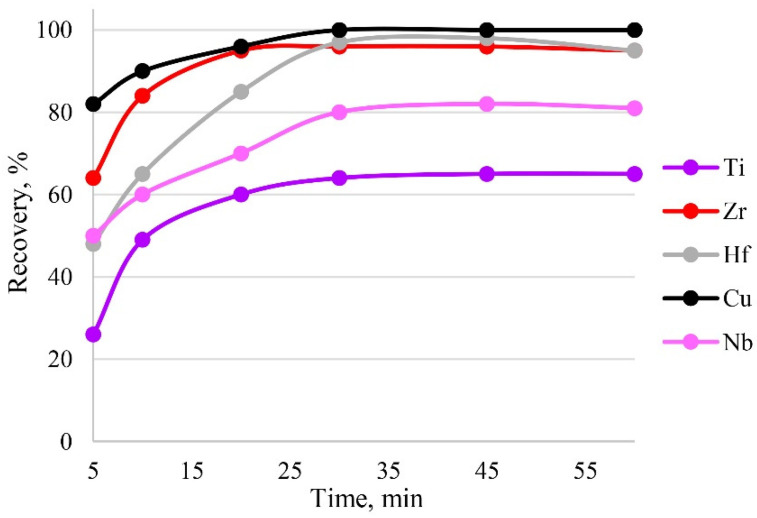
The dependence of recovery on time. Sorption conditions: 0.1 mol/L HCl, C_EDTA/NH3_ = 50/2500 mg/L, m_PTE_ = 0.1 g, V _liquid phase_ = 10 mL, T = 60 °C, c_analytes_ = 1 mg/L.

**Figure 10 molecules-27-05275-f010:**
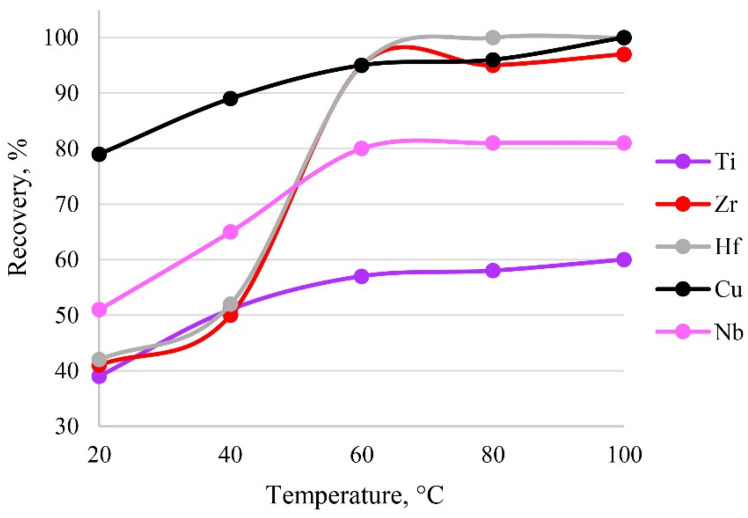
The dependence of recovery on temperature. Sorption conditions: 0.1 mol/L HCl, C_EDTA/NH3_ = 50/2500 mg/L, m_PTE_ = 0.1 g, V _liquid phase_ = 10 mL, t = 30 min, c_analytes_ = 1 mg/L.

**Figure 11 molecules-27-05275-f011:**
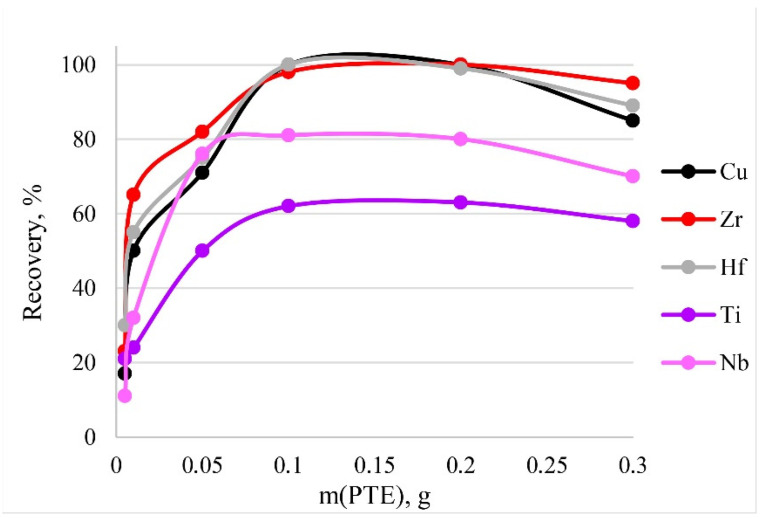
The dependence of recovery on mass of PTE. Sorption conditions: 0.1 mol/L HCl, C_EDTA/NH3_ = 50/2500 mg/L, V _liquid phase_ = 10 mL, t = 30 min, T = 60 °C, c_analytes_ = 1 mg/L.

**Table 1 molecules-27-05275-t001:** Instrumental operating conditions of ICP-OES measurements.

Forward power, W	900–1400
Coolant gas flow, L/min	15
Auxiliary gas flow, L/min	0.35
Nebuliser gas flow, L/min	0.50–1.3
Sample flow rate, rpm	60
Pump tube, mm	0.64
Radial viewing height, mm	10
Injector diameter, mm	2
Pneumatic nebuliser	SeaSpray nebuliser, glass expansion
Spray chamber	Cyclonic spray chamber, glass expansion
Wavelength (nm)	Cu 223.008, Cu 224.700, Cu 323.452, Ti 337.280, Zr 267.863, Zr 339.198, Nb 295.088, Nb 309.418, Nb 264.141, Hf 277.336, Hf 282.022

**Table 2 molecules-27-05275-t002:** Instrumental operating conditions of mass spectrometer.

Forward power/W	1300
Coolant gas flow/L·min^−1^	0.8
Auxiliary gas flow/L·min^−1^	13
Nebuliser gas flow/L·min^−1^	0.85–0.90
Sample flow rate/rpm	50
Sampling depth/relative units	101
Potential at the extractor lens/V	−400
Spray booth temperature/°C	3
Level of oxide ions/%	<2
Level of doubly charged ions/%	<1.5
Measurement mode	Peak hopping
Pneumatic nebuliser	SeaSpray Nebuliser, glass expansion
Spray chamber	Quartz conical, Peltier cooled
Isotopes of elements to be determined/*m/z*	^47^Ti, ^63^Cu, ^91^Zr, ^177^Hf, ^93^Nb

**Table 3 molecules-27-05275-t003:** The certified concentrations of the RMs (m_RM_ = 0.05 g, V_solution_ = 10 mL).

Element	RM 1 (mg/L)	RM 2 (mg/L)	RM 3 (mg/L)	RM 4 (mg/L)
Cu	0.10 ± 0.01	0.50 ± 0.02	0.49 ± 0.02	2.41 ± 0.09
Zr	0.07 ± 0.01	0.15 ± 0.02	0.040 ± 0.01	1.01 ± 0.05
Hf	0.25 ± 0.02	0.10 ± 0.01	0.27 ± 0.02	0.29 ± 0.02

**Table 4 molecules-27-05275-t004:** The recovery of Ti, Cu, Zr, Nb and Hf from SmCo-solutions, %. Sorption conditions: without acid, m_sorbent_ = 0.1 g, V _liquid phase_ = 10 mL, t = 1 h, T = 60 °C, C_analytes_ = 1 mg/L.

Element	PTE	PED	TDA	PhED
Ti	7	19	17	16
Cu	56	76	63	61
Zr	14	20	17	18
Nb	25	36	38	27
Hf	20	22	23	21

**Table 5 molecules-27-05275-t005:** The effect of acids, recovery, %. Sorption conditions: 0.1 mol/L HCl/HNO_3_, m_sorbent_ = 0.1 g, V_liquid phase_ = 10 mL, t = 1 h, T = 60 °C, C_analytes_ = 1 mg/L.

Element	HCl				HNO_3_			
	PTE	PED	TDA	PhED	PTE	PED	TDA	PhED
Ti	35	24	37	30	14	21	26	20
Cu	91	86	80	61	63	80	74	65
Zr	40	31	27	14	18	28	24	21
Nb	36	35	42	21	28	38	40	31
Hf	29	24	30	17	23	31	28	27

**Table 6 molecules-27-05275-t006:** Effect of matrix on recovery. Sorption conditions: 0.1 mol/L HCl, C_EDTA/NH3_ = 50/2500 mg/L, m_PTE_ = 0.1 g, V _liquid phase_ = 10 mL, t = 30 min, T = 60 °C, c_analytes_ = 1 mg/L.

Element	Recovery from the Co Matrix, %	Recovery from the Sm Matrix, %
Ti	0	3
Cu	87	70
Nb	61	60
Zr	18	97
Hf	11	95

**Table 7 molecules-27-05275-t007:** Analytical results and recovery data for analytes in waste SmCo reference materials. The determination of analytes was carried out via ICP-OES with preconcentration. Sorption conditions: 0.1 mol/L HCl, C_EDTA/NH3_ = 50/2500 mg/L, m_PTE_ = 0.1 g, V _liquid phase_ = 10 mL, time = 30 min, T = 60 °C, c_analytes_ = 1 mg/L.

Element	RM 1				RM 2	
	Added (mg/L)	Found (mg/L)	Recovery (%)	Added (mg/L)	Found (mg/L)	Recovery (%)
Cu	0	0.12 ± 0.01	-	0	0.48 ± 0.03	-
	0.5	0.65 ± 0.04	106	0.5	0.99 ± 0.08	102
Zr	0	0.09 ± 0.01	-	0	0.16 ± 0.01	-
	0.5	0.58 ± 0.04	98	0.5	0.68 ± 0.04	94
Hf	0	0.22 ± 0.02	-	0	0.07 ± 0.01	-
	0.5	0.70 ± 0.06	96	0.5	0.59 ± 0.03	92

**Table 8 molecules-27-05275-t008:** Analytical results for analytes, using ICP-OES with preconcentration and ICP-MS (n = 2, P = 0.95). Sorption conditions: 0.1 mol/L HCl, C_EDTA/NH3_ = 50/2500 mg/L, m_PTE_ = 0.1 g, V_liquid phase_ = 10 mL, time = 30 min, T = 60 °C, c_analytes_ = 1 mg/L.

Element	RM 3 (μg/g)				RM 4 (μg/g)			
	ICP-OES with Preconcentration	r	I(TO)	ICP-MS	ICP-OES with Preconcentration	r	I(TO)	ICP-MS
Cu	0.53 ± 0.04	0.056	0.067	0.48 ± 0.03	2.50 ± 0.14	0.197	0.237	2.33 ± 0.13
Zr	0.038 ± 0.012	0.016	0.020	0.035 ± 0.009	1.05 ± 0.08	0.113	0.135	0.99 ± 0.07
Hf	0.24 ± 0.03	0.042	0.050	0.26 ± 0.02	0.25 ± 0.03	0.042	0.050	0.30 ± 0.03

## Data Availability

All data used to support the findings of this study are included within the article.

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
