# Peer review of "Sorption Preconcentration and Analytical Determination of Cu, Zr and Hf in Waste Samarium–Cobalt Magnet Samples"

_molecules, 2022, doi:10.3390/molecules27165275_

Round 1

Reviewer 1 Report

Unfortunately, I do not recommend this manuscript for its publication in the present form. It should be revised from a chemical point of view in order to increase its scientific level.

Comments to be addressed

 - Introduction is weak in content. It should address questions such as “ Why do you want to eliminate or determine these trace impurities? What is the objective of your studies? There are more metals apart from these five you have studied?

-           - It would be also interesting to make a revision in the literature about sorption methods already used for this purpose. What are the advantages/novelty of your work in relation with others found?

-          - Why are N- and S- containing compounds interesting for metal sorption? Explain the mechanism.

-          - You should name the sorbent compounds in the text the first time they appear (what is the meaning of PTE, PED, TDA, etc?)

-        -   In the tables of sections 2.2, 2.3, figure 2, etc., you should specify the concentrations of HCl, metal ions, additional reagents, etc. It is very important to describe the experiments conditions in the table header and figure captions. Check this comment throughout the manuscript.

-        - Why did you choose these particular “additional reagents” and not others? What is the reaction with the sorbents? Did you study the effect of the concentration of the additional reagents (not only EDTA and NH3) on metal recovery?

-        -   Which is the sorbent you refer in Figure 3? Please specify.

-        -  It should be of great information to indicate de pH value of each solution, since sorption mechanisms highly depend on this variable.

-         - Do you stir the solution during experiments? Specify. If not, don’t you think it could increase the results?

-         -  What is the linear range of application of this method for each metal? Authors say “the linear range of calibration was over four orders of magnitude…” Specify.

-         -  VERY IMPORTANT: What are the certified concentrations of the Reference Materials you have used? You should compare your results with these certified values. Specify these values in tables 4 and 5.

-        -  In table 4, what is the technique used to determine these concentration? Specify in the table header.  Are RM1 and RM2 reference materials or “magnet samples” as you say?

-        -  Where were the magnet samples obtained from?

-        -  About sorption procedure: How do you heat the samples? Using a bath? Add and explain this in apparatus section.

-       -   Is all the deposit completely washed away from the filter (line 235)?

-        -   In conclusion section, you should specify the sorbents studied and the maximum final % obtained for each metal ion (with +/- SD).

-         - You should add the study of the method reproducibility and express the mean value +/- SD.

Author Response

Point «English language»: English language and style are fine/minor spell check required

Response «English language»: The revised text was checked and corrected by a native English speaker

Point «Comments and Suggestions for Authors»: Unfortunately, I do not recommend this manuscript for its publication in the present form. It should be revised from a chemical point of view in order to increase its scientific level.

Response «Comments and Suggestions for Authors»: The Manuscript improved from a chemical point of view in accordance with the proposals of the Reviewers

Point 1: Introduction is weak in content. It should address questions such as “ Why do you want to eliminate or determine these trace impurities? What is the objective of your studies? There are more metals apart from these five you have studied?

Response 1: The introduction improved with the proposals of reviewers.

Below are answers to the questions:

«Why do you want to eliminate or determine these trace impurities?»

«The waste SmCo magnets contain Sm and Co as the main components, as well as Fe, non-ferrous metals and rare metals, including Cu, Zr, Nb, Ti, Hf. Determination of these impurities is also relevant in terms of their secondary use, in particular for microelectronics».

«What is the objective of your studies?»

One of the universal techniques of the elements preconcentration in such a complex, multicomponent and non-standard object, as secondary metal-containing raw materials, containing rare metals, is a sorption preconcentration based on S- and N–containing sorbents.  As the review of published articles has shown, sorption methods were not used for waste Sm-Co magnets. The objective of this work was to investigate the capabilities of S- and N–containing sorbents for the sorption of rare and non-ferrous metals (Ti, Cu, Zr, Nb and Hf) from waste Sm-Co magnets for further ICP-OES analysis.

«There are more metals apart from these five you have studied?»

«The waste SmCo magnets contain Sm, Co, Fe, non-ferrous metals and rare metals, including Cu, Zr, Nb, Ti, Hf. Determination of these elements is relevant in terms of their secondary use. In order to improve the possibilities of the direct ICP-OES analysis the sorption preconcentration technique was used.

Point 2: It would be also interesting to make a revision in the literature about sorption methods already used for this purpose. What are the advantages/novelty of your work in relation with others found?

Response 2: The Introduction provides the information about the experience of using S- and N–containing sorbents for preconcentration of different elements in secondary raw materials. A review of publications showed that S- and N–containing sorbents have not previously been used for RE-based magnets and their waste.

In this work, for the first time, a sorption preconcentration method was developed for the determination of rare and non-ferrous metals (Cu, Zr, Nb, Ti, Hf ) by ICP-OES with high sensitivity and acceptable accuracy.

Point 3: - Why are N- and S- containing compounds interesting for metal sorption? Explain the mechanism.

Response 3: N- and S- containing  sorbents have shown their effectiveness in preconcentration of different elements in the analysis of secondary raw materials [

  1. Filatova D. G., Arkhipenko A. A., Statkus M. A., Es’kina V. V., Baranovskaya V. B., Karpov Yu. A. Sorption of Se(IV) from Aqueous Solutions with Subsequent Determination by X-Ray Fluorescence Analysis. Inorganic Materials 2021, 57, 14, 1427–1430. DOI: 10.1134/S0020168521140053.
  2. Eskina V.V., Dalnova O.A., Kareva E.N., Baranovskaya V.B., Karpov Y.A. Determination of impurities in high-purity niobium (V) oxide by high-resolution continuum source graphite furnace atomic absorption spectrometry after sorption preconcentration. Journal of Analytical Chemistry 2017, 72, 6, 649-655
  3. NEW COMPLEXING POLYMER AMINOTHIOETHER SORBENTS IN THE ANALYTICAL CONTROL OF RECYCLABLE METAL-CONTAINING RAW MATERIAL OF RARE AND NOBLE METALS. Dal’nova O.A., Dal’nova Y.S., Baranovskaya V.B., Karpov Y.A.Journal of Analytical Chemistry. 2018. Т. 73. № 3. С. 221-227.
  4. SEPARATION AND PRECONCENTRATION OF PLATINUM-GROUP METALS FROM SPENT AUTOCATALYSTS SOLUTIONS USING A HETERO-POLYMERIC S, N-CONTAINING SORBENT AND DETERMINATION BY HIGH-RESOLUTION CONTINUUM SOURCE GRAPHITE FURNACE ATOMIC ABSORPTION SPECTROMETRY. Eskina V.V., Baranovskaya V.B., Karpov Y.A., Dalnova O.A., Filatova D.G. Talanta. 2016. Т. 159. С. 103-110.
  5. Dalnova O.A., Baranovskaya V.B., Dalnova Y.S., Karpov Y.A. New Complexing Polymer Aminothioether Sorbents in the Analytical Control of Recyclable Metal-Containing Raw Material of Rare and Noble Metals. Journal of Analytical Chemistry 2018, 73, 3, 221-227. DOI: 10.1134/S1061934818030036]

These sorbents do not require desorption of elements, are easily dissolved in mineral acids, do not have a an  interfering effect during further ICP-MS and ICP-AES analysis.

The investigation of the properties of these sorbents showed that the sorption of metals by aminothioether sorbents is accelerated due to the rapid formation of donor–acceptor bonds between the nitrogen atom and the atom of the extracted element. Later, the intramolecular transfer of the metal atom takes place in the resulting complex with the fixing of this metal atom at the sulfur atom. The free nitrogen atom is again bound to the next metal atom and also transfers it to the second sulfur atom of the dithiasine sorbent ring, which explains the unusually high sorption capacity of the aminothioether sorbents with respect to a number of elements.

1.Петрухин О.М., Нефедов В.И., Золотов Ю.А. Сорбция платиновых металлов полимерным ти

оэфиром // Журн. аналит. химии. 1983. Т. 37. No 2. С. 250.

  1. Афонин М.В., Симанова С.А., Бурмистрова Н.М., Панина Н.С., Карпов Ю.А., Дальнова О.А. Сорбционное извлечение хлорокомплексов платины (II) и платины (IV) гетероцепным серосодер- жащим сорбентом // Журн. прикл. химии. 2008. Т. 81. No 11. С. 1816. (Afonin M.V., Simanova S.A., Burmistrova N.M., Panina N.S., Karpov Yu.A., Dal’nova O.A. Sorption recovery of platinum(II) and platinum(IV) chloro complexes with a heterochain sulfur–containing sorbent // Russ. J. Appl. Chem. 2008. V. 81. No 11. P. 1933.)

Point 4: You should name the sorbent compounds in the text the first time they appear (what is the meaning of PTE, PED, TDA, etc?)

Response 4: We changed «Materials and Methods» and «Results and Discussion» in places, since the experimental part contains the decoding of all sorbents.

Point 5: In the tables of sections 2.2, 2.3, figure 2, etc., you should specify the concentrations of HCl, metal ions, additional reagents, etc. It is very important to describe the experiments conditions in the table header and figure captions. Check this comment throughout the manuscript.

Response 5: Agree with the reviewer. Corrections have been made.

Point 6: Why did you choose these particular “additional reagents” and not others? What is the reaction with the sorbents? Did you study the effect of the concentration of the additional reagents (not only EDTA and NH3) on metal recovery?

Response 6: According to the literature data, when a metal interacts with a sorbent, these additional reagents are embedded in free places, making the complex more stable and increasing recovery.

Point 7: Which is the sorbent you refer in Figure 3? Please specify.

Response 7: Added to captions to all tables and figures.

Point 8: It should be of great information to indicate de pH value of each solution, since sorption mechanisms highly depend on this variable

Response 8: We believe that acidity values are more understandable than pH. pH values are given in the Manuscript (Section 3.6. The effect of sample acidity).

Point 9: Do you stir the solution during experiments? Specify. If not, don’t you think it could increase the results?

Response 9: Indicated in «Materials and Methods» (section 2.5)

Point 10: What is the linear range of application of this method for each metal? Authors say “the linear range of calibration was over four orders of magnitude…” Specify.

Response 10: ICP-OES has wide dynamic range. The linear range of calibration was over four orders of magnitude, with a correlation coefficient in excess of 0.998. Rectilinear calibration lines were confirmed for 0.01 to 10 mg/ L of minor additives, Ti, Cu, Nb, Zr and Hf.

Point 11: VERY IMPORTANT: What are the certified concentrations of the Reference Materials you have used? You should compare your results with these certified values. Specify these values in tables 4 and 5.

Response 11: The Certified concentrations of the Reference Materials are Indicated in the Table 3 (Section 2.2. Reagents and materials)

Point 12: In table 4, what is the technique used to determine these concentration? Specify in the table header.  Are RM1 and RM2 reference materials or “magnet samples” as you say?

Response 12: The text was corrected

Point 13: Where were the magnet samples obtained from?

Response 13: «Materials and Methods» says (section 2.2): «The reference materials (RM) of waste magnets (RM 1, RM 2, RM 3, RM 4) were developed at the Federal State Research and Development Institute of Rare Metal Industry “Giredmet” (Russia). The chemical compositions of RM were determined using ICP-MS».

Point 14: About sorption procedure: How do you heat the samples? Using a bath? Add and explain this in apparatus section

Response 14: Added to sections 2.1 and 2.4

Point 15: Is all the deposit completely washed away from the filter (line 235)?

Response 15: All the deposit completely is washed away from the filter. This confirms the added-found method (table 7).

In addition, after washing away all the deposit completely, the filter was washed with concentrated nitric acid. ICP-OES analysis showed that the concentration of analytes on the filter is below the limitы of detection

The addition explanation of the procedure is added to the Section «Materials and Methods»

Point 16: In conclusion section, you should specify the sorbents studied and the maximum final % obtained for each metal ion (with +/- SD).

Response 16: Added in conclusion section

Point 17: You should add the study of the method reproducibility and express the mean value +/- SD.

Response 17: This method was developed for in-house use. Therefore, according to ISO 5725, it was not necessary to study the method reproducibility. But we investigated inter-laboratory precision (I(TO)) and repeatability (repeatability limit r). The calculation was carried out in accordance with ISO 5725. The necessary data was added to the text

Author Response

Response to Reviewer 2 Comments

Point «English language»: Extensive editing of English language and style required.

Response «English language»: The revised text was checked and corrected by a native English speaker

Point «Comments and Suggestions for Authors»: Authors in manuscript “Sorption preconcentration and analytical determination of Ti, Cu, Zr, Nb and Hf in waste samarium-cobalt magnet samples for quality control of recycling schemes” proposed the extraction of Ti, Cu, Zr, Nb and Hf using PED, TDA, PhED and PTE sorbents. The determination of analytes was carried out by ICP-OES. Authors also studied the effects of: addition of reagents and its concentration, acidity, temperature, matrix, etc. The theme of the manuscript could be interesting, but the presentation of the results needs huge changes. In my opinion manuscript in this form should not be accepted for publication. Now, the manuscript seems to be a report from studies. Introduction, references and experimental descriptions are very poor. The language of the manuscript need also correction.

Response «Comments and Suggestions for Authors»: The Manuscript improved from a chemical point of view in accordance with the proposals of the reviewers

Point 1: Abstract - Authors should provide explanation to the abbreviations.

Response 1: Agree with the reviewer. Corrections have been made

Point 2: Experimental part - Synthesis of sorbent is not described. There is no information about quantities of added reagent like EDTA, etc. There is no information why the HCl was used for the acidity test. There is no information why the mixture of sorbent and sample was heated?

Response 2: We have described the synthesis of sorbents (section 2.3).

The information about quantities of added reagents (EDTA, etc.) is provided in the table headers and figure captions.

Hydrochloric acid is used to dissolve magnets and form a stable metal cations for further bonding with the sorbent. (Section 3.3)

The mixture of sorbent and sample was heated to increase the recovery. This effect is shown in the figure 7.

Point 3: Descriptions of Tables and Figures - Most of descriptions are incorrect

Response 3: «Descriptions of Tables and Figures» improved in accordance with the proposal of the reviewer

Point 4: - Results and discussion - This whole paragraph needs the biggest corrections. Interpretations of obtained results are very poor.

Response 4: « Results and discussion » improved in accordance with the proposal of the reviewer

.

Point 5: Figure 2 - Should be presented as a column graph.

Response 5: The data is presented as a column graph

Point 6: Figure 3b - On the graph there is no results for the higher concentration of ammonia.

Response 6: We haven’t done experiments with the higher concentration of ammonia, because high concentration of ammonia can influence on the  ICP-OES results and on the introduction system of spectrometer.

Point 7: Line 138 “high temperature leads to the destruction of the functional groups”. Any proves??

Response 7: The additional explanation is added to Section 3.8

Reviewer 3 Report

Comments on Molecules-1831925

Major comments:

The main focus of this study is on “Sorption preconcentration and analytical determination of Ti, Cu, Zr, Nb and Hf in waste samarium-cobalt magnet samples 3 for quality control of recycling scheme”.

The scope and main findings of the research are not well presented. Research gaps and novelty of the study with the previous knowledge is not incorporated that is confusing for the scope of the study since it is already available piles of scientific documents. The results shown in this study are lack of innovation and novelty. Discussion on figures/tables is presented in a very poor way. In general, the results are not critically justified nor properly compared with other studies to justify the conclusions made.

It is obvious the quality of the manuscript does not meet the standards of molecules, therefore should be rejected in its present form or needs major revision.

Further specific comments are given below for the revision of the manuscript thoroughly.

Specific comments:

1.                Abstract: Some of the data found in abstract cannot be found in the manuscript or is not well presented there.

2.                  Title: is more promising than the work suggests. Where are the recycling schemes?

3.                  Introduction: Avoid lumping references. Instead, summaries the main contribution of each referenced paper in a separate sentence. The last pharagraph in the introduction part should contain the aim of the work and novelty and not the main result!

4.                  The introduction needs to be more emphasized on the research work with a detailed explanation of the whole process considering past, present and future scope. It needs to be more emphasized in the research work with a detailed explanation of the whole process. Research gaps should be highlighted more clearly and future applications of this study should be added.

5.                  The novelty of the study is not presented!

6.                  All abbreviations mentioned for the first time should be defined.

7.                  Please clarify the abbreviation of polythioether sorbent! Authors sometimes use the abbreviation as PET and sometimes as PTE through writing the manuscript.

8.                  Figures:  are not in a consistent form. Please uniform!

9.                  For FTIR spectra analysis authors have no cited reference? Should be confirmed with similar research!

10.              Section 2.3. What was the concentration of the acids used? Should be added in Table 2.

11.              Section 2.4. What was the concentration of reagents used to assist the extraction of analytes from sorbent?

12.              Section 2.5. Authos should state that the performance of the effect of reagent concentration was done for PTE sorbent. Should also be added in Figure 3 caption. There is no discussion in regard to the extraction of Ti and Nb.

13.              Section 2.7 and 2.8 If the general sorption procedure is applied why were different values in Figure 5 and 6 for same paremeters shown? When testing time, the temperature was 60 °C (optimal was 30 min) and when testing the temperature for same value (60 °C) different results were presented. Should they not be same?

14.              Why authors did not perform the optimization of the effect of sorbent mass and analyte concentration as a key preconcentration process factors?

15.              Section 2.9: What did the authors consider as an interfering agent? What is the tolerance of the difference in recovery? The results of obtained recovery values were compered with what situation?

16.              Section 2.10: What was the reason that authors choose a solution with the concentration of analyte of 0.8 mg/L for relative standard determination?

17.              Section 2.11: The accuracy was tested for Cu, Zr and Hf, why not for Ti and Nb, since the title cover all 5 analytes? Please elaborate for which solution are obtained data given in table 5!

18.              Section 3.1: What are the conditions used to obtain FTIR spectra and for ICP-MS? Please mention.

19.              FTIR analysis is not is not enough for characterization of synthesized sorbents. Additional analysis should be employed.

20.              Results and discussion section: The obtained values in the results are just stated in the text without explaining them. Explain the reasons behind your trends/values and discuss them critically with literature.

21.              Authors are advised to uniform the unit for concentration in whole manuscript  (mg/L instead M) as well as Table 4: Uniform Reference M1, RM2….

22.              Pay attention to superscripts and subscripts! Please carefully go through the entire manuscript and correct it.

Author Response

Response to Reviewer 3 Comments

Point «English language»: Moderate English changes required

Response «English language»: The revised text was checked and corrected by a native English speaker

Point «Comments and Suggestions for Authors»: The main focus of this study is on “Sorption preconcentration and analytical determination of Ti, Cu, Zr, Nb and Hf in waste samarium-cobalt magnet samples 3 for quality control of recycling scheme”.

The scope and main findings of the research are not well presented. Research gaps and novelty of the study with the previous knowledge is not incorporated that is confusing for the scope of the study since it is already available piles of scientific documents. The results shown in this study are lack of innovation and novelty. Discussion on figures/tables is presented in a very poor way. In general, the results are not critically justified nor properly compared with other studies to justify the conclusions made.

It is obvious the quality of the manuscript does not meet the standards of molecules, therefore should be rejected in its present form or needs major revision.

Further specific comments are given below for the revision of the manuscript thoroughly.

Response «Comments and Suggestions for Authors»: The Manuscript improved from a chemical point of view in accordance with the proposals of the reviewers

Point 1: Abstract: Some of the data found in abstract cannot be found in the manuscript or is not well presented there.

Response 1: The text corrected

Point 2: Title: is more promising than the work suggests. Where are the recycling schemes?

Response 2: The title was corrected according with the Reviewer Comment

Point 3: Introduction: Avoid lumping references. Instead, summaries the main contribution of each referenced paper in a separate sentence. The last pharagraph in the introduction part should contain the aim of the work and novelty and not the main result!

Response 3: The Introduction was revised according to the reviewer’s comments.

Point 4: The introduction needs to be more emphasized on the research work with a detailed explanation of the whole process considering past, present and future scope. It needs to be more emphasized in the research work with a detailed explanation of the whole process. Research gaps should be highlighted more clearly and future applications of this study should be added.

Response 4: The Introduction was revised according to the reviewer’s comments.

Point 5: The novelty of the study is not presented!

Response 5: A review of publications showed that S- and N–containing sorbents have not previously been used for RE-based magnets and their waste. In this work, for the first time, a sorption preconcentration method was investigated for the determination of rare and non-ferrous metals (Cu, Zr, Nb, Ti, Hf ) by ICP-OES with high sensitivity and acceptable accuracy.

Point 6: All abbreviations mentioned for the first time should be defined.

Response 6: The abbreviations were defined.

Point 7: Please clarify the abbreviation of polythioether sorbent! Authors sometimes use the abbreviation as PET and sometimes as PTE through writing the manuscript.

Response 7: The text corrected

Point 8: Figures:  are not in a consistent form. Please uniform!

Response 8: The Figures were corrected.

Point 9: For FTIR spectra analysis authors have no cited reference? Should be confirmed with similar research!

Response 9: The references are given in the Section 3.1. Characterisation of sorbents

Point 10: Section 2.3. What was the concentration of the acids used? Should be added in Table 2.

Response 10: We have added sorption conditions in the headers of all tables and figures.

Point 11: Section 2.4. What was the concentration of reagents used to assist the extraction of analytes from sorbent?

Response 11: Sorption conditions were added to the headers of all tables and figures.

Point 12: Section 2.5. Authos should state that the performance of the effect of reagent concentration was done for PTE sorbent. Should also be added in Figure 3 caption. There is no discussion in regard to the extraction of Ti and Nb.

Response 12: This information was added to the Section 3.5. We have added information about Nb and Ti to this section

Point 13: Section 2.7 and 2.8 If the general sorption procedure is applied why were different values in Figure 5 and 6 for same paremeters shown? When testing time, the temperature was 60 °C (optimal was 30 min) and when testing the temperature for same value (60 °C) different results were presented. Should they not be same?

Response 13: As we understood, the comment concerns the difference between empirical values. As can be seen from the Figures 9 and 10, the differences are not significant and correspond to the uncertainties of the results.

Point 14: Why authors did not perform the optimization of the effect of sorbent mass and analyte concentration as a key preconcentration process factors?

Response 14: The results of the effect of sorbent mass were given in the Section 3.9.

According experimental data, the maximum extractable concentration of analytes is 7 mg/L. With a further increase of analyte concentration, the recovery will decrease. This information is added to the Section 3.11.

Point 15: Section 2.9: What did the authors consider as an interfering agent? What is the tolerance of the difference in recovery? The results of obtained recovery values were compered with what situation?

Response 15: The interfering influence of matrix elements is presented in the Section 3.10. The matrix effect.

It was found that Sm affected the extraction of Cu and that Co had an effect on the extraction of Ti, Zr, Hf.

Point 16: Section 2.10: What was the reason that authors choose a solution with the concentration of analyte of 0.8 mg/L for relative standard determination?

Response 16: The data in the Section 3.11 is given as an example. RSDs for other concentrations are presented in tables 7 and 8.

Point 17: Section 2.11: The accuracy was tested for Cu, Zr and Hf, why not for Ti and Nb, since the title cover all 5 analytes? Please elaborate for which solution are obtained data given in table 5!

Response 17: Information of analyzed solutions are given in Tables 7 and 8.

The maximum recovery of Ti and Nb, using S- and N-containing sorbents in investigated conditions were 60±5 % and 82±3%, respectively.   Therefore, there is no data for Ti and Nb in the tables 7 and 8.

Point 18: Section 3.1: What are the conditions used to obtain FTIR spectra and for ICP-MS? Please mention.

Response 18: The conditions of ICP-MS and FT-IR analysis described in the Section 2.1. Apparatus

Point 19: FTIR analysis is not is not enough for characterization of synthesized sorbents. Additional analysis should be employed.

Response 19: Additional analysis (NMR analysis) is specified in the Section 3.1. Characterisation of sorbents.

Point 20: Results and discussion section: The obtained values in the results are just stated in the text without explaining them. Explain the reasons behind your trends/values and discuss them critically with literature.

Response 20: The Section « Results and discussion » was improved in accordance with the proposals of the reviewer

Point 21: Authors are advised to uniform the unit for concentration in whole manuscript  (mg/L instead M) as well as Table 4: Uniform Reference M1, RM2….

Response 21: The text corrected

Point 22: Authors are advised to uniform the unit for concentration in whole manuscript  (mg/L instead M) as well as Table 4: Uniform Reference M1, RM2….

Response 22: The text corrected

Round 2

Reviewer 3 Report

Authos have made a lot of corrections and incorporated all reviewers suggestions therefore suggest  to accept this paper for publication.

Author Response

The authors thank the Reviewer for careful attention to the manuscript and acception the paper for publication.
